# Fast and Accurate Registration of Terrestrial Point Clouds Using a Planar Approximation of Roof Features

Maria Alicandro [ID], Luca Di Angelo *[ID], Paolo Di Stefano [ID], Donatella Dominici [ID], Emanuele Guardiani [ID] and Sara Zollini [ID]

Heritechne Center, University of L'Aquila, Piazzale E. Pontieri, Monteluco di Roio, 67100 L'Aquila, Italy; maria.alicandro@univaq.it (M.A.); paolo.distefano@univaq.it (P.D.S.); donatella.dominici@univaq.it (D.D.); emanuele.guardiani@univaq.it (E.G.); sara.zollini@univaq.it (S.Z.)
* Correspondence: luca.diangelo@univaq.it

**Abstract:** 3D reconstructed models are becoming more diffused daily, especially in the Cultural Heritage field. These geometric models are typically obtained from elaborating a 3D point cloud. A significant limit in using these methods is the realignment of different point clouds acquired from different acquisitions, particularly for those whose dimensions are millions of points. Although several methodologies have tried to propose a solution for this necessity, none of these seems to solve definitively the problems related to the realignment of large point clouds. This paper presents a new and innovative procedure for the fine registration of large point clouds. The method performs an alignment by using planar approximations of roof features, taking the roof's extension into account. It looks particularly suitable for the alignment of large point clouds acquired in urban and archaeological environments. The proposed methodology is compared in terms of accuracy and time with a standard photogrammetric reconstruction based on Ground Control Points (GCPs) and other ones, aligned by the Iterative Closest Point method (ICP) and markers. The results evidence the excellent performance of the methodology, which could represent an alternative for aligning extensive photogrammetric reconstructions without the use of GCPs.

**Keywords:** point cloud registration; multi-UAV scanning registration; shape features recognition; particle swarm optimization

## 1. Introduction and State of the Art

Image-based and laser scanning are the mainly employed techniques for the 3D reconstruction of extensive areas. The first one, in particular, has been growing incredibly in the last few years, thanks to the possibility of being combined with the use of Unmanned Aerial Vehicles (UAVs), or drones. These can be equipped with a professional camera and can survey large and open spaces in a short time, with a low effort and, especially, a minimal cost. Laser scanning, on the other side, requires a higher effort to be used, so it is preferred in situations where a UAV cannot operate, such as in the 3D reconstruction of a large enclosed environment.

One of the current significant limitations in using these technologies is managing the extensive data set resulting from the acquisitions. Nowadays, large-scale 3D scanners can reach a few centimeters and even millimeters of resolution. Millions of points can characterize the resulting 3D point cloud when capturing large area environments. Such a large point cloud is quite hard to process with the current technologies [1], in particular for three different aspects, which are:

- Post-processing operations and tessellation for generating the 3D mesh;
- Storage of information;
- Alignment with other point clouds.

This last point, in particular, is becoming crucial in the last few years because, the characteristics of 3D scanners being complementary, the research community has advocated the integration of point clouds coming from different acquisition modalities [2]. This integration, being in most cases the point clouds defined in different coordinate systems, requires a well-known registration procedure that allows for aligning multiple geometric models to a unique reference frame. One of the most used methodologies for point cloud alignment is based on the use of markers. At least three markers are appropriately positioned on the area to be scanned. The point clouds are aligned by imposing the overlap of the recognized markers. The main limit of this methodology is the precision of the result, which also depends on the resolution of the digital images used to acquire the markers. Furthermore, the markers have to be positioned in an area to be detected in both acquisitions. This is not possible in the case of acquisitions performed a long time apart. In such cases, the use of natural markers is preferred, which, on the other side, is more difficult to detect accurately.

Most of the other existing procedures for point cloud alignment employ a coarse-to-fine alignment [3]. The coarse registration is addressed to establish a rough alignment between the point clouds. This step is fundamental for the methodologies in use for the fine registration, which are very sensitive to the initial alignment in terms of computational cost and final result. The coarse registration can adopt several strategies: Aiger et al. [4], for example, introduced the *Four-points congruent set*, a method allowing for extracting a set of four co-planar points whose intersectional diagonal ratios are invariant. This property is used to verify the matching between two point clouds and evaluate the alignment's rigid transformation (rotation and translation). The weakness of this method is the high computational cost required to verify the matches between point sets.

Other methods treat the coarse alignment by using a *probabilistic approach*, searching for one-to-many correspondences between density functions. Jian et al. [5] modeled the point clouds through the Gaussian mixture models. Golyanik et al. and Zang et al. [6,7] proposed an improved and refined probabilistic registration framework. However, the limit of these methodologies is the incapability to work with large point clouds because registration results strictly depend on the sampling point clouds.

Finally, a last class of methodologies is based on *deep learning techniques*. Their significant advantage consists in the capability to recognize automatically, with good performance, features in the point cloud, reducing the subjectivity introduced by the operators when they perform these operations manually. The identified features are then used to establish the transformation matrix necessary for the alignment of point clouds. Some of the most interesting and recent works in this area have been presented by Qi et al. [8], Deng et al. [9,10], Yang et al. [11], and Wang et al. [12]. Their methods allow the direct alignment of unordered point sets. However, their reliability is limited only to small-scale indoor environments.

The previously presented methodologies do not allow for obtaining, so far, a satisfying alignment in metrological terms. Therefore, they are typically applied only as a first step of a more complex registration procedure that foresees, in the second part, the use of *Iterative Closest Point* (ICP) or *Normal Distribution Transform* (NDT), with their respective variants.

The ICP is a diffused method for fine registration of point clouds: it is quite simple to implement and the results are satisfying, especially when a good initial alignment is provided between the point clouds. ICP [13] performsoptimal registration by conversely resolving the nearest point-to-point correspondence and the optimal rigid transformation until convergence [14]. ICP searches for one-to-one correspondences between the points; this generates difficulties when the point clouds are characterized by heterogeneous densities, a typical drawback of 3D scanner technologies. ICP is also very sensitive to outliers, noise, and occlusions and misbehaves when the point clouds to be aligned have been acquired in different conditions (presence of cars, fauna, different light or weather conditions). Furthermore, this method is onerously computational since hundreds of thousands of points must be processed in a mathematical optimization process. Consequently, several variants of the classical ICP method have been proposed during these

years. Sharp et al. [15], for example, suggested the use of invariant features combined with the idea of the geometric distance to minimize the effects of noise. Bae et al. [16] introduced a reviewed ICP, which uses curvature and normal vectors to identify the correspondences. Bouaziz et al. [17] presented a technique that treats the ICP as a sparse optimization problem whose solution minimizes the effects of outliers. In addition to the quality of the point cloud alignment, the computational efficiency of the method is another critical aspect. Uhlenbrock et al. [18] proposed a 2D array based k-d tree to speed up the iterative process. Pavlov et al. [19] introduced the Anderson acceleration technique in ICP, helping to reduce the number of iterations required for the method to converge. Nevertheless, the sensitivity to noise and outliers and the low efficiency of the method remain an open topic for the ICP.

NTD [20,21] is another method employed for the fine alignment. Its working principle is similar to the coarse alignment's previously mentioned probabilistic approach. The point cloud is represented through a Gaussian distribution; each distribution is characterized by a specific Probability Density Function (PDF). Assigned two-point clouds to align, the NTD determines, through a nonlinear optimization problem, the transformation which minimizes the dissimilarities between the two PDFs. As opposed to the ICP, the NTD can also operate with low-density point clouds. On the other side, this method is conceptually more difficult to implement; the definition of PDFs requires a voxelization of the point cloud space, and the results of the nonlinear optimization are strictly dependent on the first alignment tentative.

The point clouds registration performed by using geometric features is a promising strategy to reduce the computational complexity of the ICP method, which can achieve high-quality results. This approach for the point cloud alignment is based on the overlap of properly selected geometric features recognized in the common area of each point cloud. This approach is, in general, much more efficient than the ICP since the alignment process is performed by using a limited amount of data, those associated with few geometric entities, generally planes. Furthermore, it performs high-quality results not so far from those obtained using ICP. Stamos et al. [22] presented a method where the alignment of point clouds is based on the intersection lines of the geometric planes associated with the vertical facets of buildings. A similar methodology was proposed by Yang et al. [23]. Some methods look interesting since they improve the efficiency of the feature alignment process. Dold et al. [24] introduced a technology that identifies the 3D planar patches of the mesh and combines them by using a constrained search technique so that the matching combinations are reduced. Xu et al. [25] proposed an automatic strategy for aligning planar patches based on the voxelization of the point cloud. In each voxel, an implicit plane representation is adopted to fit the surface of points. An eigenvalue decomposition evaluates the quality of the approximating plane with the surface.

Wu and Fan in [26] present an approach for registering airborne LiDAR point clouds based on matching corresponding linear plane features. First, the point clouds of the building roofs are extracted from the two LiDAR datasets using the method proposed in [27]. By an iterative process, the points used to approximate the roofs are those for which the residual error compared to the approximating plane is less than a threshold value. Then, the normal vectors are calculated for every simple roof facet of the corresponding buildings. The vectors of the roof facets are used to establish an observation equation system to estimate the transformation between the two datasets; the correspondence between roofs is manually detected. The least-squares method is used to solve the observation system with redundant feature-pairs. The registration is performed in two steps: first, the rotation matrix calculation and then the 3D translation vector computation. The results show that the method does not consider the size of the roofs in the optimization but only their number. This leads to incorrect registration in the case of a non-uniform distribution of roofs.

Rabbani et al. [28] illustrated a methodology that uses other kinds of features, such as cylinders, spheres, and planes that are extracted directly from the point cloud.

Registration performed using geometric features is fascinating because these, especially those extended concerning the point cloud's dimensions, have low sensitivity to

noise and outliers. Moreover, when enough geometric primitives are recognized in the point cloud, this one can be set aside for the successive alignment operations because the geometric features are defined by all the geometric attributes necessary for the alignment.

All the described methodologies using geometric features for the alignment have been tested in urban and interior environments since, in these cases, all the acquired objects are defined by planar features. In this paper, the geometric feature-based approach is used to align point clouds acquired in a vast natural environment with few diffused artificial elements such as groups of houses or historical environments. In particular, the possibility of associating planar features with roofs, which represent isolated elements in the environment and are characterized by very irregular surfaces, will be investigated. In addition, a new representation of the ideal flat feature associated with the roofs is proposed considering the roofs' size in the observation equation system. This allows for an optimization that better considers the spatial distribution of roofs and not simply their spatial position and orientation.

In the first part of the paper, ad-hoc experimentation that has been led to show the validity and the repeatability of approximating roofs by planes will be presented. The second part will discuss an application of the proposed methodology to the test case of Alba Fucens, Italy, where the roofs of some ancient buildings have been used to merge two separate surveys. The alignment results have been compared with the "ground truth" thanks to some Ground Control Points (GCPs), uniformly distributed into the aligned point clouds. The presented methodology results have also been compared with those obtained from an ICP and a marker-based alignment.

## 2. The Proposed Methodology

The problem solved by the presented methodology consists of the alignment of two surveys of territory having a shared zone, where geometric features can approximate some anthropic artifacts. The methodology here proposed uses a typical element recurrent in the anthropized environments: the roofs of buildings. The roof pitches are elements to which ideal geometric features could be associated (planes) and then used as a reference for the alignment procedure of two distinct surveys. The proposed computer-aided approach starts with two sets of unordered point clouds to align, $\wp_A^c$ and $\wp_B^c$, representing two different areas, $A$ and $B$; the problem to be solved requires that:

$$A \cap B = C \neq \varnothing \tag{1}$$

In each point cloud, the points of the $C$ area are segmented so that the $C$ area is described by the survey of the region $A$ $\{\wp_{A \subset C}\}$ and the survey of the region $B$ $\{\wp_{B \subset C}\}$. The presence of roof elements (non-ideal features) or other geometric entities to be associated with ideal features is required in $C$. In what follows, only roof pitches will be considered. The set of roof features in $C$ must be identified as a set of non-ideal features with which ideal geometric elements can be associated (ideal-features). The ideal features are used to determine the two surveys' alignment uniquely.

The methodology introduced in this paper allows for obtaining the fine alignment of multiple point clouds, two clouds at a time, by three key phases:

- non-ideal feature identification;
- ideal-features association;
- registration of the sets of ideal features and transformation matrix evaluation for the alignment of the point clouds.

### 2.1. Non Ideal-Feature Identification

The proposed methodology uses some of the roofs in the territory acquired as non-ideal features, with which ideal features (planes) are associated, leading to the alignment process. Roofs are identifiable in every urban environment and, thanks to their area extension, are high-weight elements for driving an alignment process. The set of non-ideal features (roofs)

needs to be selected, in number and orientation, so that all the degrees of freedom necessary for alignment of the point clouds are constrained. Some methods described in the literature can be used for this purpose to automatize the identification process. Fan et al. in [27] present an approach for roof facet segmentation based on ridge detection and hierarchical decomposition along ridges. The process starts with detecting 3D points of roof ridges as the local maximum height. For each detected roof ridge, 3D points are segmented to their corresponding roof facets being located on the same plane as the seed points. The process terminates when no roof ridges can be detected. Dahaghin et al. in [29] proposed a method to extract building roofs from the visible point cloud. The method includes ground filtering, vegetation removal, wall removal based on geometric properties, and, finally, segmentation of remaining points. The roof segmentation is performed by a modified version of the connected component labeling algorithm proposed by Lumialn et al. in [30] that divides the selected point cloud into smaller parts separated by the minimum distance condition, and each part forms a connected component. Awrangjeb et al. in [31] propose an automatic 3D roof extraction method by integrating LIDAR (Light Detection In addition, Ranging) data and multispectral ortho imagery. Using the ground height from a DEM (Digital Elevation Model), the raw LIDAR points are separated into the ground and non-ground points. The latter are segmented using the image line guided segmentation technique using the ground mask and color and texture information from the orthoimagery to extract the roof planes. These automatic methods presented in the literature show excellent results in roof segmentation. Since this paper aims to analyze mainly the association and registration phases, each roof pitch (non-ideal feature) is manually selected and segmented by an operator from the point clouds to which it belongs, so that $h$-th non-ideal features are identified, one for each roof pitch. The resulting patches, $\{\Gamma_1^A, \Gamma_2^A, \ldots, \Gamma_h^A$ for $\wp_A; \Gamma_1^B, \Gamma_2^B, \ldots, \Gamma_h^B$ for $\wp_B\}$, are the real features associated with the $h$ roofs identifiable in the point set $C$.

### 2.2. Ideal-Features Association

An ideal planar feature $\{\Pi_j^i\}$ is associated with each identified patch $\{\Gamma_y^x\}$. The repeatability of the ideal feature association to the roof patch is an important aspect that must be evaluated. Roofs are not planar objects, but they are composed of tiles that can be different in dimensions, colors, shapes, and integrity (cf. Figure 1). The continuity of roofs can be broken by chimney pots, antennas, and windows, making the planar approximation even more complex. Each time the same roof is surveyed, a different point cloud is obtained, but the ideal features associated with it have to represent the same ideal geometric entity with a high level of repeatability.

Specific experimentation has been carried out to validate the present hypothesis. The roofs characterizing a private building sited in Genzano (42°21′09.7″N, 13°19′29.8″E), L'Aquila, Italy, have been captured through 20 UAV surveys. The UAV surveys have been performed by the use of a DJI Mavic Pro, whose technical specifications are indicated in Table 1. The missions have been accomplished between 2 July and 5 July 2021, with good weather but different light conditions. Figure 2 shows a top view of the building subjected to the experimentation, while the flight plan parameters are reported by Table 2.

Approximately 40 photos were captured during each survey. These have been imported into Agisoft Metashape Pro (v. 1.7.1) as separated chunks, where they have been processed to obtain a 3D mesh using the Ultra High preset. The resulting meshes have been aligned with each other by defining four Control Points, recognizable thanks to the use of artificial markers that an operator has placed on the rooftop. The alignment has been performed into Agisoft Metashape using the integrated Chunk Alignment tool. Then, the meshes have been exported into Cloud Compare and manually segmented to isolate the parts of roofs. Figure 3a shows the results of the alignment and segmentation process for one of the roof features identified in the acquired building (evidenced by yellow in Figure 2). In particular, a roof feature is identified by those surfaces having the same normal, with a specific tolerance value, without interruptions. Figure 3a evidences a not negligible dispersion in the overlapping between the aligned meshes.

**Table 1.** Technical specification of the instrumentation used for the UAV photogrammetry surveys in Genzano.

| | | |
|---|---|---|
| **Sensor** | Camera | DJI FC220 |
| | Resolution | 12 Mpixel |
| | Focal length | 26 mm |
| | Sensor dimensions | 1/2.3″ (CMOS) |
| | FoV | 78 deg |
| **UAV** | Typology | Micro UAV Quadricopter |
| | Brand | DJI |
| | Model | Mavic Pro |
| | Weight at takeoff | 734 g |
| | Autonomy | 27 min |
| | Operating altitude | 5000 m |

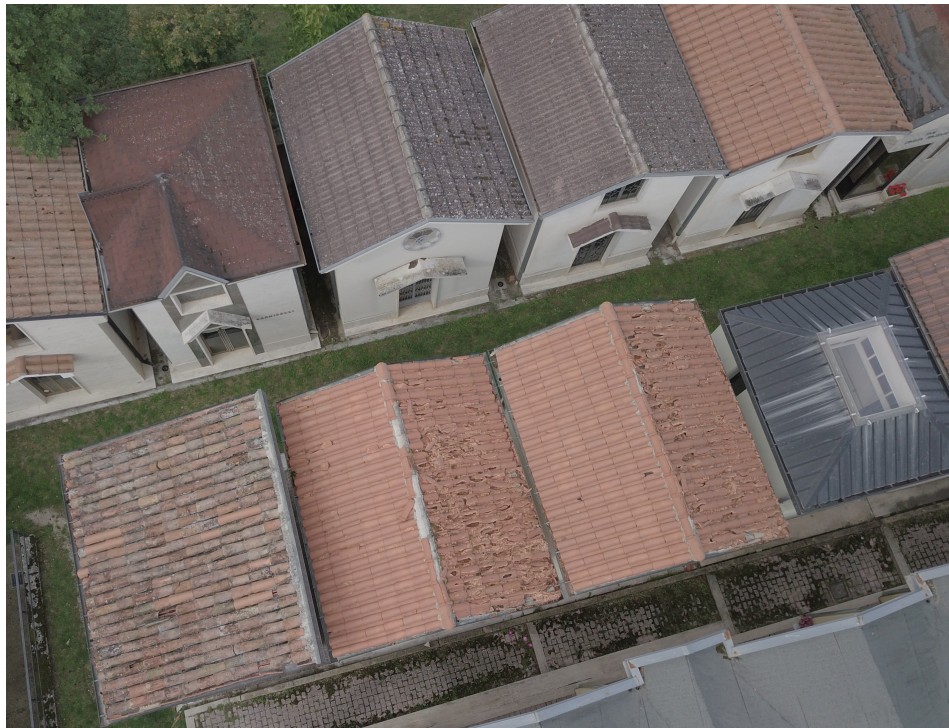

**Figure 1.** An example of roof tiles characterized by different shape, dimensions, colors and integrity.

**Table 2.** Adopted configuration for the 20 surveys.

| | |
|---|---|
| Camera angle | 80 deg |
| Front overlap | 80% |
| Side overlap | 70% |
| Flight altitude | 23 m |
| Ground Simple Distance | 1.3 cm/px |

A periodic function describes the roof contour, neglecting the outliers, whose period remains constant while the magnitude changes. These dispersions are for several reasons. A systematic source of error is introduced by the changes in light conditions, which lead to a different evaluation of roof contours by the photogrammetric reconstruction algorithm. This problem, described by Aber et al. [32], can be identifiable from Figure 3b. A source of casual error is introduced by the capturing sensors of the UAV camera. Finally, the discretization from the tessellation process introduces other differences among the points clouds. The

present experimentation shows that the approximating planes have limited variability, despite the differences between the obtained models from the different acquisitions.

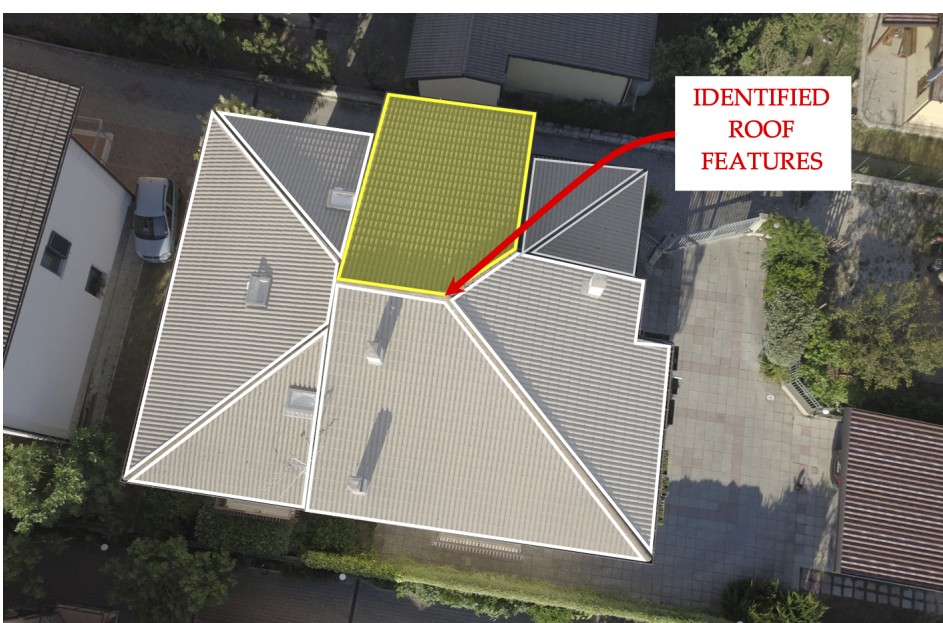

**Figure 2.** Top view of the building chosen for experimenting the approximation of roofs by planes.

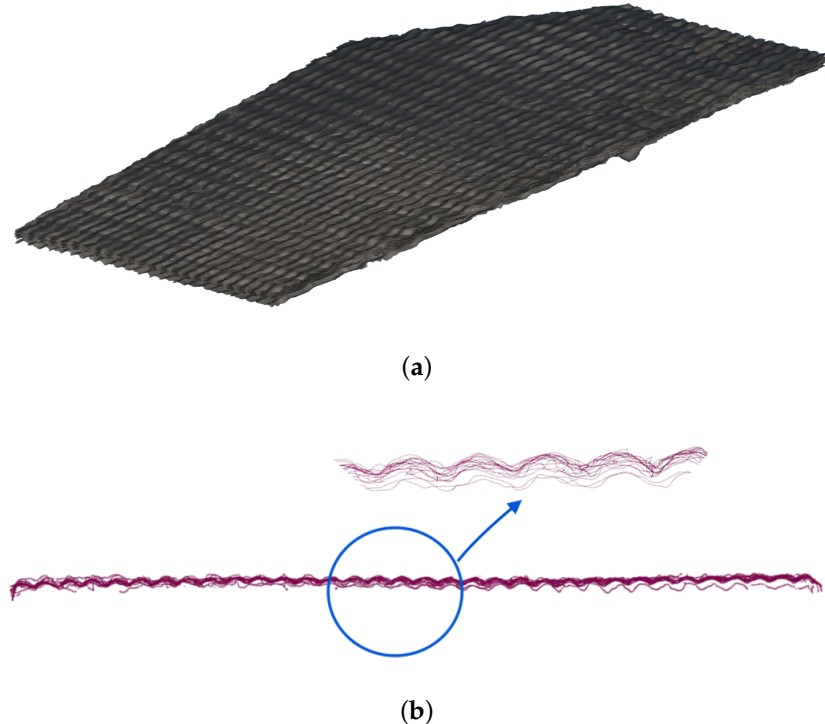

(**a**)

(**b**)

**Figure 3.** Results of alignment and segmentation of one of the roof features of the acquired building. (**a**) View of the aligned mesh for one of the recognized roof features; (**b**) 2D contours obtained from the cross section of the roof feature recognized in (**a**).

For each one of the recognized roof features of the acquired building, the approximating planes according to $L_1$, $L_2$, and $L_{inf}$ norms have been evaluated. Being the implicit equation of a plane is as follows:

$$ax + by + cz + d = 0 \qquad (2)$$

the values of *a*, *b*, *c* and *d* have been determined for the metrics previous mentioned. Only the results of the roof feature shown in Figure 3 have been reported in Table 3, those associated with the other recognized roof features being very similar to this in terms of statistical behaviour.

**Table 3.** Values of coefficients *a*, *b*, *c* and *d* for the 20 meshes of the roof feature identified by Figure 3.

| Acquisition | $L_1$ | | | | $L_2$ | | | | $L_{inf}$ | | | |
|---|---|---|---|---|---|---|---|---|---|---|---|---|
| | *a* [-] | *b* [-] | *c* [-] | *d* [m] | *a* [-] | *b* [-] | *c* [-] | *d* [m] | *a* [-] | *b* [-] | *c* [-] | *d* [m] |
| 1 | $-2.24 \times 10^{-4}$ | $-8.89 \times 10^{-3}$ | 1.00 | $-1.30 \times 10^{-2}$ | $-1.26 \times 10^{-3}$ | $-8.25 \times 10^{-3}$ | 1.00 | $-9.68 \times 10^{-3}$ | $9.35 \times 10^{-4}$ | $-9.56 \times 10^{-3}$ | 1.00 | $-1.66 \times 10^{-2}$ |
| 2 | $-2.66 \times 10^{-4}$ | $-8.80 \times 10^{-3}$ | 1.00 | $-9.64 \times 10^{-3}$ | $-1.20 \times 10^{-3}$ | $-8.04 \times 10^{-3}$ | 1.00 | $-7.47 \times 10^{-3}$ | $6.83 \times 10^{-4}$ | $-9.53 \times 10^{-3}$ | 1.00 | $-1.22 \times 10^{-2}$ |
| 3 | $2.39 \times 10^{-3}$ | $-1.14 \times 10^{-2}$ | 1.00 | $-1.80 \times 10^{-3}$ | $1.10 \times 10^{-3}$ | $-1.05 \times 10^{-2}$ | 1.00 | $2.13 \times 10^{-3}$ | $3.83 \times 10^{-3}$ | $-1.27 \times 10^{-2}$ | 1.00 | $-5.66 \times 10^{-3}$ |
| 4 | $-1.58 \times 10^{-3}$ | $-8.35 \times 10^{-3}$ | 1.00 | $-1.76 \times 10^{-3}$ | $-2.36 \times 10^{-3}$ | $-8.18 \times 10^{-3}$ | 1.00 | $2.03 \times 10^{-3}$ | $-6.44 \times 10^{-4}$ | $-8.79 \times 10^{-3}$ | 1.00 | $-5.04 \times 10^{-3}$ |
| 5 | $1.79 \times 10^{-3}$ | $-1.02 \times 10^{-2}$ | 1.00 | $-7.66 \times 10^{-3}$ | $1.06 \times 10^{-3}$ | $-9.09 \times 10^{-3}$ | 1.00 | $-6.35 \times 10^{-3}$ | $2.35 \times 10^{-3}$ | $-1.14 \times 10^{-2}$ | 1.00 | $-7.84 \times 10^{-3}$ |
| 6 | $2.15 \times 10^{-3}$ | $-6.12 \times 10^{-3}$ | 1.00 | $-1.58 \times 10^{-2}$ | $1.30 \times 10^{-3}$ | $-5.79 \times 10^{-3}$ | 1.00 | $-1.33 \times 10^{-2}$ | $3.02 \times 10^{-3}$ | $-6.65 \times 10^{-3}$ | 1.00 | $-1.80 \times 10^{-2}$ |
| 7 | $7.30 \times 10^{-4}$ | $-3.17 \times 10^{-3}$ | 1.00 | $-2.40 \times 10^{-2}$ | $-8.29 \times 10^{-4}$ | $-6.86 \times 10^{-3}$ | 1.00 | $-1.00 \times 10^{-2}$ | $9.62 \times 10^{-4}$ | $-6.96 \times 10^{-3}$ | 1.00 | $-1.71 \times 10^{-2}$ |
| 8 | $2.20 \times 10^{-3}$ | $-9.92 \times 10^{-3}$ | 1.00 | $-8.32 \times 10^{-3}$ | $1.34 \times 10^{-3}$ | $-9.25 \times 10^{-3}$ | 1.00 | $-6.10 \times 10^{-3}$ | $3.06 \times 10^{-3}$ | $-1.08 \times 10^{-2}$ | 1.00 | $-1.02 \times 10^{-2}$ |
| 9 | $3.01 \times 10^{-3}$ | $-1.15 \times 10^{-2}$ | 1.00 | $4.06 \times 10^{-3}$ | $2.02 \times 10^{-3}$ | $-1.12 \times 10^{-2}$ | 1.00 | $7.72 \times 10^{-3}$ | $3.90 \times 10^{-3}$ | $-1.18 \times 10^{-2}$ | 1.00 | $1.19 \times 10^{-3}$ |
| 10 | $-1.07 \times 10^{-3}$ | $-5.71 \times 10^{-3}$ | 1.00 | $-1.43 \times 10^{-2}$ | $-1.72 \times 10^{-3}$ | $-3.76 \times 10^{-3}$ | 1.00 | $-1.57 \times 10^{-2}$ | $-3.95 \times 10^{-4}$ | $-5.95 \times 10^{-3}$ | 1.00 | $-1.65 \times 10^{-2}$ |
| 11 | $-7.59 \times 10^{-4}$ | $-8.77 \times 10^{-3}$ | 1.00 | $4.65 \times 10^{-3}$ | $-1.34 \times 10^{-3}$ | $-8.50 \times 10^{-3}$ | 1.00 | $6.04 \times 10^{-3}$ | $-1.83 \times 10^{-4}$ | $-8.77 \times 10^{-3}$ | 1.00 | $2.47 \times 10^{-3}$ |
| 12 | $1.29 \times 10^{-2}$ | $-1.03 \times 10^{-2}$ | 1.00 | $3.22 \times 10^{-3}$ | $1.16 \times 10^{-2}$ | $-9.46 \times 10^{-3}$ | 1.00 | $6.69 \times 10^{-3}$ | $1.41 \times 10^{-2}$ | $-1.13 \times 10^{-2}$ | 1.00 | $3.79 \times 10^{-4}$ |
| 13 | $-2.16 \times 10^{-4}$ | $-9.41 \times 10^{-3}$ | 1.00 | $-4.44 \times 10^{-3}$ | $-1.14 \times 10^{-3}$ | $-8.58 \times 10^{-3}$ | 1.00 | $-2.07 \times 10^{-3}$ | $9.47 \times 10^{-4}$ | $-1.04 \times 10^{-2}$ | 1.00 | $-7.76 \times 10^{-3}$ |
| 14 | $3.01 \times 10^{-3}$ | $-1.15 \times 10^{-2}$ | 1.00 | $4.06 \times 10^{-3}$ | $2.02 \times 10^{-3}$ | $-1.12 \times 10^{-2}$ | 1.00 | $7.72 \times 10^{-3}$ | $3.90 \times 10^{-3}$ | $-1.18 \times 10^{-2}$ | 1.00 | $1.19 \times 10^{-3}$ |
| 15 | $-1.37 \times 10^{-3}$ | $-6.16 \times 10^{-3}$ | 1.00 | $-1.21 \times 10^{-2}$ | $-1.74 \times 10^{-3}$ | $-6.85 \times 10^{-4}$ | 1.00 | $-2.25 \times 10^{-2}$ | $-2.87 \times 10^{-4}$ | $-4.30 \times 10^{-3}$ | 1.00 | $-1.99 \times 10^{-2}$ |
| 16 | $8.74 \times 10^{-4}$ | $-8.33 \times 10^{-3}$ | 1.00 | $-3.91 \times 10^{-3}$ | $1.05 \times 10^{-4}$ | $-7.69 \times 10^{-3}$ | 1.00 | $-1.49 \times 10^{-3}$ | $1.68 \times 10^{-3}$ | $-9.23 \times 10^{-3}$ | 1.00 | $-5.80 \times 10^{-3}$ |
| 17 | $1.94 \times 10^{-4}$ | $-7.72 \times 10^{-3}$ | 1.00 | $2.40 \times 10^{-3}$ | $-2.58 \times 10^{-4}$ | $-8.02 \times 10^{-3}$ | 1.00 | $5.28 \times 10^{-3}$ | $5.52 \times 10^{-4}$ | $-7.51 \times 10^{-3}$ | 1.00 | $3.26 \times 10^{-4}$ |
| 18 | $-1.70 \times 10^{-3}$ | $-1.17 \times 10^{-2}$ | 1.00 | $6.85 \times 10^{-3}$ | $-2.33 \times 10^{-3}$ | $-1.15 \times 10^{-2}$ | 1.00 | $9.49 \times 10^{-3}$ | $-9.29 \times 10^{-4}$ | $-1.21 \times 10^{-2}$ | 1.00 | $4.13 \times 10^{-3}$ |
| 19 | $-9.07 \times 10^{-3}$ | $-2.22 \times 10^{-2}$ | 1.00 | $6.89 \times 10^{-2}$ | $-8.60 \times 10^{-3}$ | $-2.03 \times 10^{-2}$ | 1.00 | $6.06 \times 10^{-2}$ | $-9.74 \times 10^{-3}$ | $-2.43 \times 10^{-2}$ | 1.00 | $7.86 \times 10^{-2}$ |
| 20 | $3.64 \times 10^{-3}$ | $-1.40 \times 10^{-2}$ | 1.00 | $-1.30 \times 10^{-2}$ | $2.72 \times 10^{-3}$ | $-1.45 \times 10^{-2}$ | 1.00 | $-7.96 \times 10^{-3}$ | $4.47 \times 10^{-3}$ | $-1.35 \times 10^{-2}$ | 1.00 | $-1.71 \times 10^{-2}$ |

The obtained values have been statistically analysed to determine the repeatability and reliability of approximating roofs by planes. The results of the analysis have been reported in Table 4 and lead to the following considerations: each one of the adopted norms for the plane approximation shows good results, although $L_2$ evidence better behavior with respect to the other ones. The standard deviations of the coefficients *a*,*b* and *c* are the same for $L_1$, $L_2$ and $L_{inf}$. These refer to the cosine directors of the plane, and the obtained standard deviations can be translated into minimal orientation errors. The orientation error among the obtained planes being minimal, the *d* parameter can be interpreted as the distance among them: the obtained standard deviation, using $L_2$, is 1.7 cm. Since this value is comparable to the Ground Simple Distance of the UAV surveys, the approximation of roofs by planes seems robust.

**Table 4.** Statistical analysis of the coefficients *a*, *b*, *c* and *b* reported by Table 3.

| | $L_1$ | | | | $L_2$ | | | | $L_{inf}$ | | | |
|---|---|---|---|---|---|---|---|---|---|---|---|---|
| | *a* [-] | *b* [-] | *c* [-] | *d* [m] | *a* [-] | *b* [-] | *c* [-] | *d* [m] | *a* [-] | *b* [-] | *c* [-] | *d* [m] |
| Mean | 0.001 | −0.010 | 1.000 | −0.002 | 0.000 | −0.009 | 1.000 | 0.000 | 0.002 | −0.010 | 1.000 | −0.004 |
| Standard deviation | 0.004 | 0.004 | 0.000 | 0.019 | 0.004 | 0.004 | 0.000 | 0.017 | 0.004 | 0.004 | 0.000 | 0.021 |

In light of the achieved results, the $L_2$ fitting method has been used for the experimental part of the present work, described in Section 3.

### 2.3. Ideal-Features Registration

The realignment problem is solved by minimizing the distances from the set of $n_c$ ideal features $\{\Pi_A^h\}$ associated with $\{\wp_{A \subset C}\}$ with the set of corresponding ideal features $\{\Pi_B^h\}$ in $\{\wp_{B \subset C}\}$. Since the roofs are determined in the overlapping part of two points clouds, correspondences are automatically derived with an initial registration of roofs points based on the PCA method. The apex *h* ( $h = 1$ to $n_c$) identifies the corresponding ideal feature in each of the two points clouds. $\{\wp_{B \subset C}\}$ is assumed as a fixed point cloud, and $\{\wp_{A \subset C}\}$ as a moving one. Registration performs the alignment of corresponding ideal features recognized in $\{\wp_{A \subset C}\}$ and $\{\wp_{B \subset C}\}$ so that the sum of the distances of two corresponding ideal features is minimized:

$$min \sum_{h=1}^{n_c} \left\| dist(\Pi_h^A, \Pi_h^B) \right\| \tag{3}$$

For this purpose, the ideal features ($\Pi_h^B$) in the fixed point cloud are represented by their implicit equations. In contrast, those in the moving point cloud ($\Pi_h^A$) by five reference points are automatically computed from the parametric equation of the plane: the four extreme points of the ideal feature and its barycentric point ($\{\mathbf{P}_{i,h}^A\}$ $i = 1$ to $5$ and $\mathbf{P}_{i,h}^A \in \Pi_h^A$) (Figure 4).

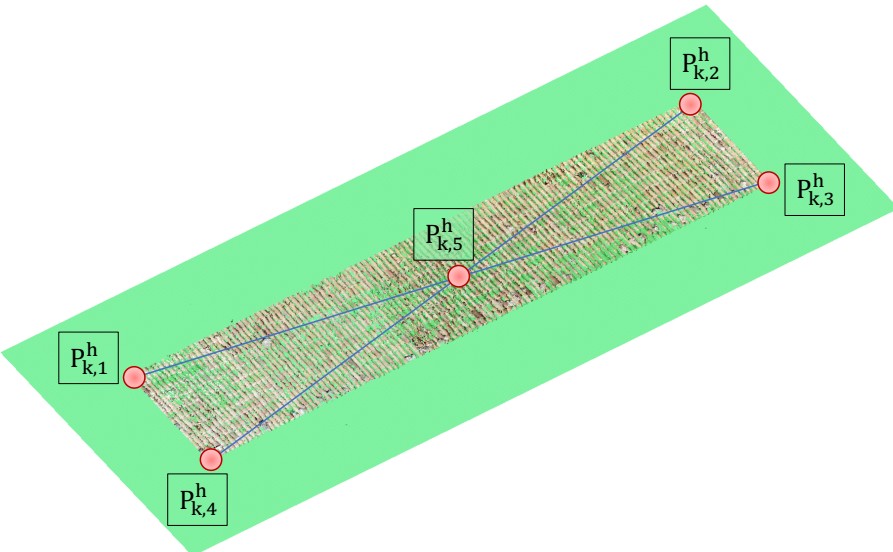

**Figure 4.** Example of the non-ideal feature with superimposed ideal-features and the reference points.

In order to evaluate the distance $dist(\Pi_h^A, \Pi_h^B)$, for each ideal feature belonging to the moving point cloud $\{\Pi_h^A\}$, the registration is achieved by an iterative method that minimizes the following objective function:

$$O(\mathbf{x})_k = \sum_{h=1}^{n_c} \sum_{i=1}^{5} \left| d([\mathbf{P}_{i,h}^A]_k, \Pi_h^B) \right|$$

where:

- $n_c$ is the number of the ideal features in each point cloud to be aligned;
- $d([\mathbf{P}_{i,h}^A]_k, \Pi_h^B)$ is the distance between the $i$-th reference point at the $k$-th step $[\mathbf{P}_{i,h}^A]_k$ and the corresponding $h$-th ideal feature $\{\Pi_h^B\}$;
- $[\mathbf{P}_{i,h}^A]_k = RM * SM * TM * [\mathbf{P}_{i,h}^A]_{k-1}$ is expressed by homogeneous coordinates;
- $RM_k = \begin{bmatrix} x(1) & x(2) & x(3) & 0 \\ -x(2) & x(4) & x(5) & 0 \\ -x(3) & -x(5) & x(6) & 0 \\ 0 & 0 & 0 & 1 \end{bmatrix}$ is the rotation matrix at the $k$-th step;
- $SM_k = \begin{bmatrix} x(7) & 0 & 0 & 0 \\ 0 & x(7) & 0 & 0 \\ 0 & 0 & x(7) & 0 \\ 0 & 0 & 0 & 1 \end{bmatrix}$ is the scale matrix at the $k$-th step;
- $TM_k = \begin{bmatrix} 1 & 1 & 0 & x(8) \\ 0 & 1 & 0 & x(9) \\ 0 & 0 & 1 & x(10) \\ 0 & 0 & 0 & 1 \end{bmatrix}$ is the translation matrix at the $k$-th step;

As the minimization strategy, a meta-heuristic algorithm is used in the proposed methodology: the Particle Swarm Optimization (PSO) [33]. PSO has been chosen since,

with respect to the exact methods, it has a lower computational burden and lack of strict usage hypothesis. Although the use of the PSO does not guarantee the convergence to a global minimum, in the application analysed here, we have the initial association between the features of the fixed point cloud with the corresponding one of the moving permitted appropriate solutions in all the considered scenarios. The result of the registration steps is the roto-translation non-rigid matrix:

$$RTM_f = RM_0 \times TM_0 \times RM \times SM \times TM \tag{4}$$

## 3. Experiments and Analysis

The methodology presented here has been tested in Alba Fucens, L'Aquila, Italy. Alba Fucens was an old Roman City founded in 303 B.C., and its ruins represent the broadest archaeological site in the Apennines. The extension of archaeological excavations and elements of interest in this city covers an area of approximately 20 hectares. Therefore, the digital 3D reconstruction of this archaeological site looks suitable for testing the proposed technique. Figure 5 shows the main phases of the experiment. The area reported by Figure 6 has been acquired, and 3D reconstructed through two different UAV surveys. These were performed on the date of 20 December 2021, using a DJI Matrice 200 V2, equipped with a DJI Zenmuse X5S, whose technical specifications are listed in Table 5.

**Table 5.** Technical specification of the instrumentation used for the UAV photogrammetry surveys in Alba Fucens.

|        |                   |                      |
| ------ | ----------------- | -------------------- |
|        | Camera            | Zenmuse X5S          |
|        | Resolution        | 20.8 Mpixel          |
| Sensor | Focal length      | 15 mm                |
|        | Sensor dimensions | 4/3″                 |
|        | Weight            | 461 g                |
|        | Typology          | UAV quadcopter       |
|        | Brand             | DJI                  |
| UAV    | Model             | Matrice 200 Series V2 |
|        | Weight at takeoff | 4.7 kg               |
|        | Autonomy          | 38 min               |
|        | Operating altitude | 3000 m              |

The flight plans have been set to guarantee a good overlap between images and establish the final GSD (Ground Sample Distance) as a function of the sensor focal length and the height flight. A side lap of 60% and an overlap of 70% have been established for these surveys. The flights have been set at 30 m of altitude from the starting area (red circle in Figure 6) with a GSD of 0.66 cm/pixel. As the area is quite steep, the roof cluster, which is the starting point, is located in the highest part of the two surveys. The lowest area achieved about 55 m of flight altitude and 1.21 cm/pixel GSD. 417 and 475 images for the first and second flights have been taken. The flight plans for the two surveys are reported in Figure 6. In addition, during the survey, 25 points were acquired by a GNSS (Global Navigation Satellite System) receiver in NRTK (Network Real-Time Kinematic) mode ($\sigma$: 2–3 cm) that will be used for geo-referencing the point clouds and assessing their quality.

The acquired images have been processed into Agisoft Metashape Pro, obtaining two different point clouds:

1. the first point cloud $\{\wp_{Ref,S1}\}$ referred to the area of Survey 1;
2. the second point cloud $\{\wp_{Ref,S2}\}$ referred to the area of Survey 2;

The elaborations have been performed using the same settings for all the point clouds and, in particular, the most advanced preset (highest) for the Image Alignment and ("ultra-high") for the Dense Point Cloud generation. Table 6 presents the estimated errors on GCPs

and CPs for every survey. An average error of 5.35 cm has been measured on the GCPs and 5.01 cm on the CPs for Survey 1, while the same error is 4.99 cm and 4.57 cm on the CPs for Survey 2. The obtained values demonstrate good quality of the final reconstructions, whose errors are comparable with those detected in similar works described by the literature. In Table 7, the data associated with the photogrammetric reconstructions have been listed. Figure 7 shows the results of Dense Point Cloud generation and the position of GCPs and CPs.

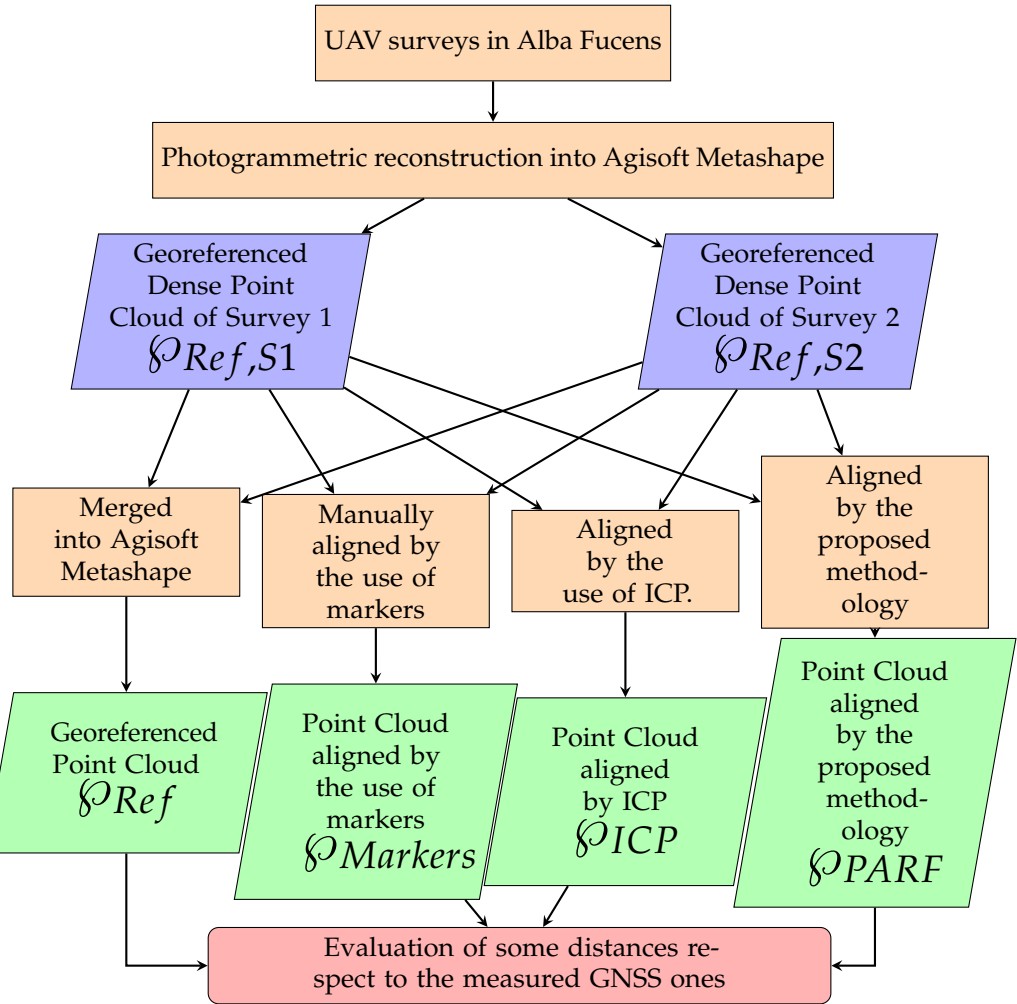

**Figure 5.** Workflow of the adopted experimentation to validate the proposed methodology.

**Table 6.** GCPs and CPs RMSE of Surveys 1 and 2.

| Survey 1 | X Error [cm] | Y Error [cm] | Z Error [cm] | Total [cm] |
| --- | --- | --- | --- | --- |
| GCPs RMSE (8) | 3.67 | 2.76 | 2.75 | 5.35 |
| CPs RMSE (4) | 1.27 | 1.76 | 4.51 | 5.01 |
| **Survey 2** | **X Error [cm]** | **Y Error [cm]** | **Z Error [cm]** | **Total [cm]** |
| GCPs RMSE (6) | 3.55 | 2.03 | 2.75 | 4.99 |
| CPs RMSE (5) | 3.46 | 2.76 | 1.11 | 4.57 |

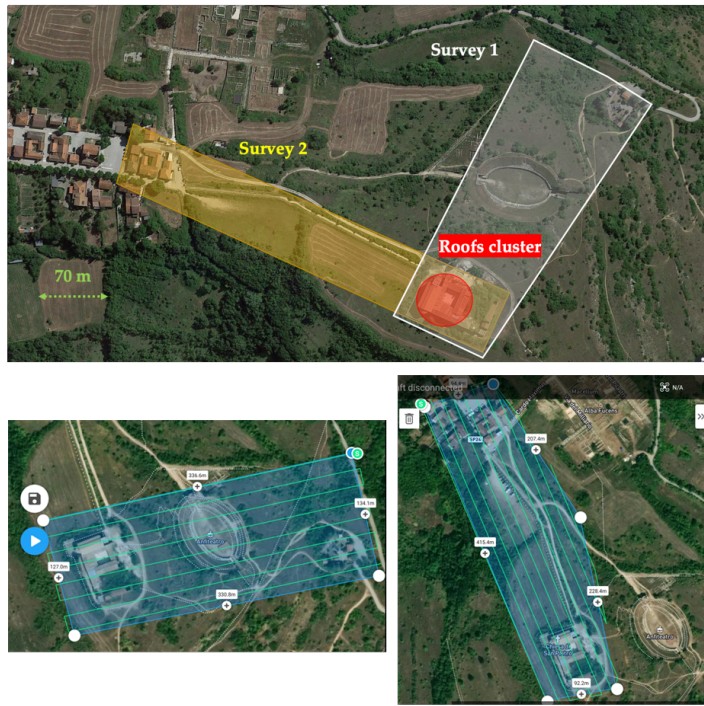

**Figure 6.** Overview of Alba Fucens and area covered by the two surveys and the flight plans of S1 and S2.

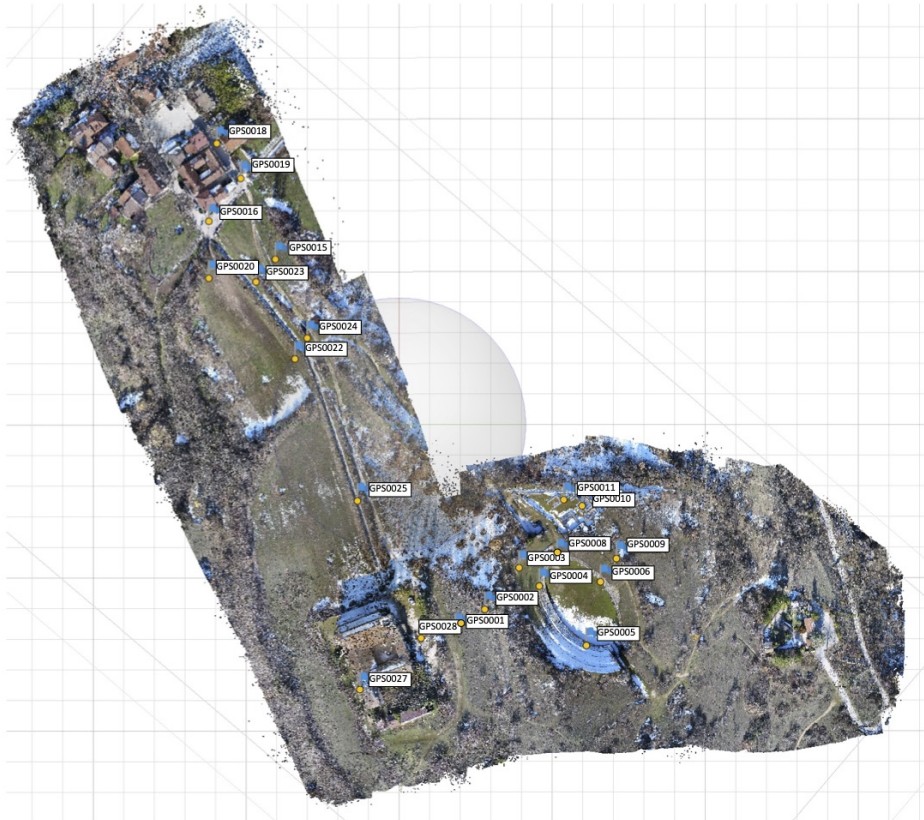

**Figure 7.** Dense Point Cloud generation into Agisoft Metashape and position of GCPs.

**Table 7.** Data related to the photogrammetric reconstructions.

|  | Survey 1 | Survey 2 |
|---|---|---|
| Ground Simple Distance | 1.2 cm/px | 1.2 cm/px |
| Photos | 417 | 475 |
| Ground Control Point and Check Points | 12 | 11 |
| Time spent for the Dense Point Cloud generation | 1 days and 5 h | 1 days and 12 h |

These two point clouds have been registered by four different methods, which are described in the following:

- $\wp_{ref} \rightarrow$ the registration of $\{\wp_{Ref,S1}\}$ and $\{\wp_{Ref,S2}\}$ has been performed into Agisoft Metashape. The two surveys, which have been processed in two different chunks, have been aligned with the "Align chunk" tool in the Program, using the marker-based method and the high preset. Then, the point clouds have been merged into a single one by the "Merge chunk" method. The resulting points cloud, characterized by over one and a half billion points, has been exported in .ply format according to the "WGS 84/UTM zone 33N" coordinate system. The GCP RMSE and CP RMSE are, respectively, 6.11 cm and 5.58 cm;

- $\wp_{PARF} \rightarrow$ the PARF registration, described in Section 2, has been coded in Matlab. In Figure 8a, the features used to perform the alignment of the two-point clouds are shown; these are represented by the roof patches of San Pietro church (cf. Figures 6 and 9). At the end of the optimization process, the objective function $O(\mathbf{x})$ assumed a value of 2.635. The evaluated transformation matrix $RTM_f$ has been applied to the point cloud $\{\wp_{Ref,S2}\}$ for the alignment with the reference point cloud $\{\wp_{Ref,S1}\}$, obtaining the $\{\wp_{PARF}\}$. In Figure 8b, the point clouds of the features of Survey 1 have been superimposed on the features of Survey 2. In the same figure, the distances' color map between the point cloud of Survey 2 to the reference feature of the Survey 1 is depicted.

- $\wp_{Markers} \rightarrow$, the registration between $\{\wp_{Ref,S1}\}$ and $\{\wp_{Ref,S2}\}$, has been performed by the use of natural markers; this is a standard practice in Reverse Engineering applications. Two-point clouds can be aligned by recognizing at least three common markers and applying a rigid transformation that permits them to overlap. This possibility is sub-ordered to the perfect correspondence between the considered point sets. This hypothesis is nearly impossible to verify in real-world situations, especially for photogrammetry. The overlapping between the point sets is generally handled as an optimization problem. The distance between the identified corresponding points is minimized; more correspondences are identified, and the quality of the realignment process should be better. Here, the "point pair realignment" procedure, implemented by CloudCompare, has been used for the marker-based alignment. The markers have been chosen manually by extracting a set of five markers identifiable in the common area of the two point clouds ($\{\wp_{Ref,S1}\} \cap \{\wp_{Ref,S2}\}$) (Figure 9). The markers have been selected on the same roofs used as a reference to apply the *PARF* alignment method. In this way, the results obtained by the two methodologies are comparable. The alignment has been performed with a Root Mean Square Error of the distance between the markers of 22 cm;

- $\wp_{ICP} \rightarrow$; another alignment has been executed by applying an ICP methodology. In this case, the points in the intersection area of Survey 1 and Survey 2 (*AcapB*) have been used to apply the ICP. Due to a large number of points in $\{\wp_{Ref,S1}\}$ and $\{\wp_{Ref,S2}\}$, a decimation has been done, resulting in a mean distance between points of 0.03 m. This value is comparable to the point cloud location estimated error. The ICP algorithm implemented in CloudCompare has been applied, assuming a RSE of $1 \times 10^{-6}$ and excluding the outliers from the optimization. The registration resulting from the ICP performs an RMSE of the distances between the points in the two clouds of 2 cm.

Six GCPs have been selected to compare the accuracy of the registration of $\wp_{Ref}$, $\wp_{Markers}$, $\wp_{ICP}$, and $\wp_{PARF}$: three belonging to Survey 1 (GPS0015, GPS0018 and GPS0020), three to the Survey 2 (GPS0008, GPS0009 and GPS0011). The Euclidean distance between these pairs of the selected markers has been calculated (cf. Figure 10) and compared with the reference one, resulting from the GNSS measurements. Table 8 reports the measured distances.

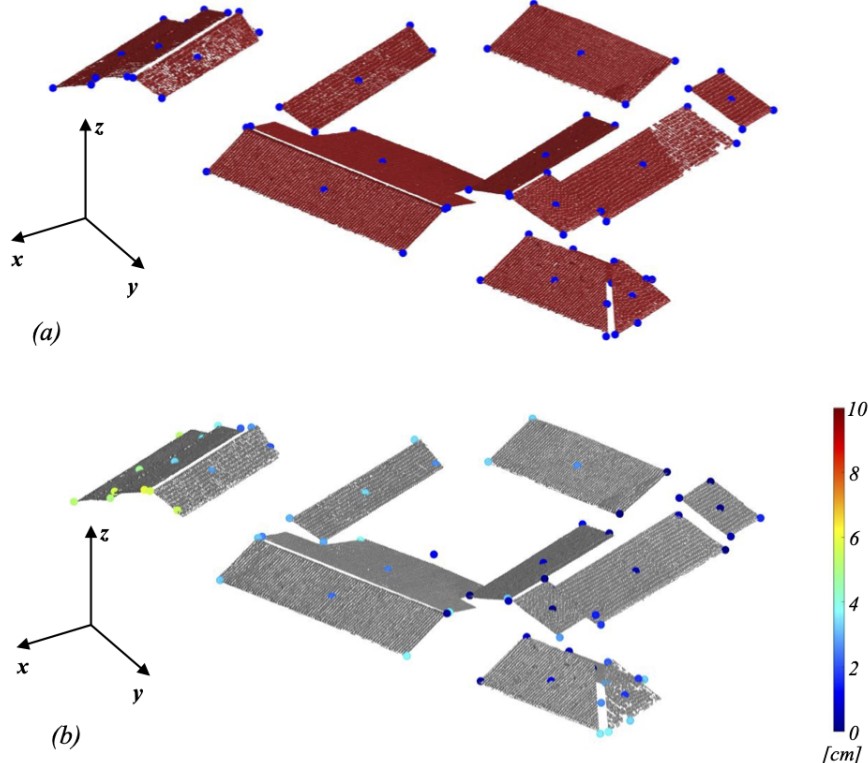

**Figure 8.** Results of registration method: (**a**) reference points superimposed on "moving" point cloud; (**b**) final position of reference points superimposed on "fixed" point cloud.

As expected, the marker method looks the worst; an average error of 0.33 m results from the three observed distances with respect to the GNSS measurement. The flaws introduced during the picking process explain such a significant inaccuracy, which is even more impacting when natural markers are used. Furthermore, due to the low dimensions of the data set utilized for alignment, local distortion significantly impacts the marker-based technique. On the other hand, the ICP method looks like the better one. An average error of 0.09 m results from the reference distances. The good ICP results can be explained by the sampling data set's size, which in this case is extended to the entire area shared by the two studies. The presence of an inaccuracy, on the other side, indicates that local distortions have a significant influence that can only be addressed by using GCPs.

The results of the presented methodology (PARF) are not so dissimilar from those obtained from the ICP alignment. An average error of 0.12 m results from the three measurements in Table 8. In this case, the sampling data set dimension is significantly smaller than the ICP. Moreover, it should be considered an additional source of error, generated by the approximation of the roof patches by planes and quantitatively experimented in Section 2.2.

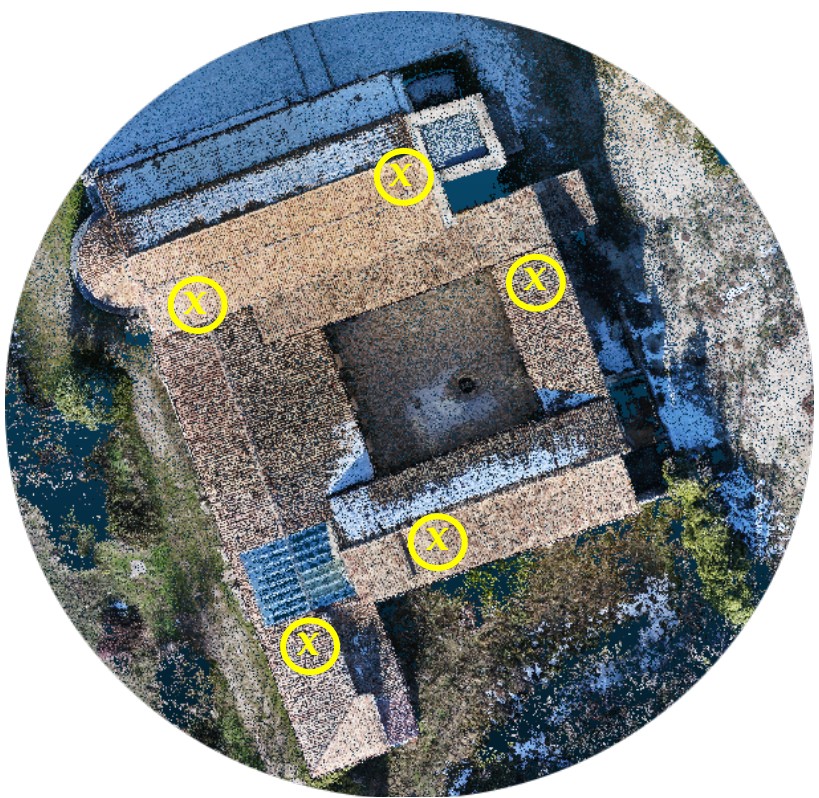

**Figure 9.** Position of markers adopted for the marker-based alignment.

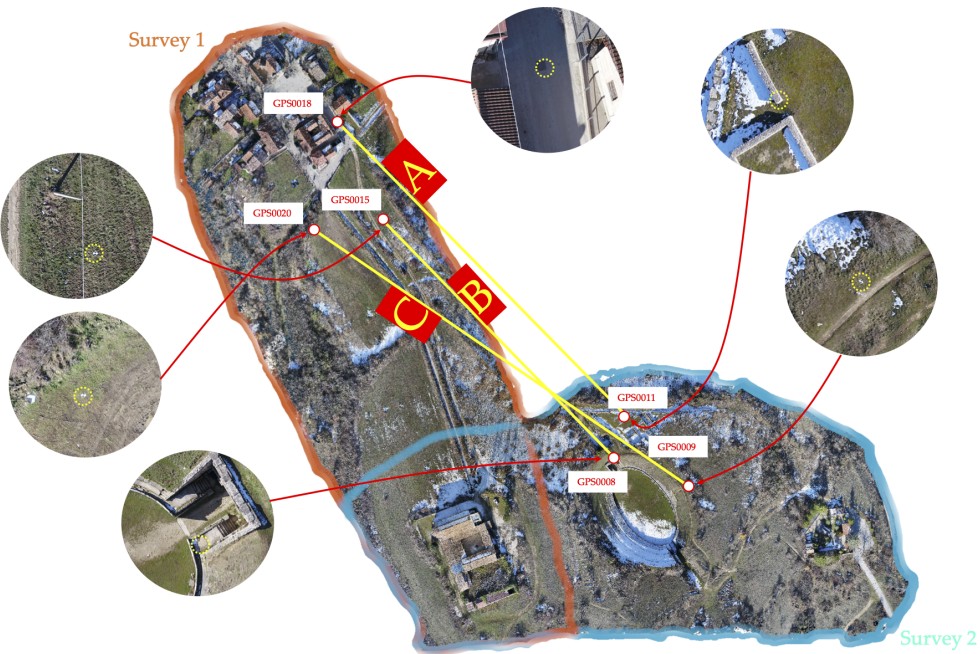

**Figure 10.** Measured distances (in yellow) between GCPs for the evaluation of the realignment process.

**Table 8.** Measured distances (cf. Figure 10) in $\wp_{Ref}$, $\wp_{Markers}$ and $\wp_{PARF}$ point clouds. * The obtained GNSS distance is calculated starting from the GNSS position in NRTK mode with $\sigma$: 0.02–0.03 m.

|  |  | A [m] | B [m] | C [m] |
|---|---|---|---|---|
| **GNSS \*** |  | **303.91** | **250.30** | **301.59** |
| $\wp_{Ref}$ |  | 303.96 | 250.32 | 301.65 |
| error | [m] | 0.05 | 0.02 | 0.06 |
|  | [%] | 0.002 | 0.001 | 0.002 |
| $\wp_{Markers}$ |  | 304.26 | 250.58 | 301.95 |
| error | [m] | 0.35 | 0.28 | 0.36 |
|  | [%] | 0.012 | 0.011 | 0.012 |
| $\wp_{ICP}$ |  | 304.00 | 250.37 | 301.70 |
| error | [m] | 0.09 | 0.07 | 0.11 |
|  | [%] | 0.003 | 0.003 | 0.004 |
| $\wp_{PARF}$ |  | 304.03 | 250.39 | 301.73 |
| error | [m] | 0.12 | 0.09 | 0.14 |
|  | [%] | 0.004 | 0.004 | 0.005 |

The decimation has provided an additional source of uncertainty during the post-processing (point-point space set to 0.03 cm). This operation has been necessary to process the point clouds into CloudCompare by the workstation used for experimentation, equipped with an Intel Xeon E5 with 64 Gb of RAM. Considering all the sources of errors introduced by the photogrammetric reconstruction and the post-processing operations, the results of the realignment by the proposed methodology look very interesting: the PARF-based method makes a maximum relative error of 0.11%, whereas that based on markers registration 0.40%. The results obtained using the proposed methodology did not require any artificial marker; the realignment process depends on the presence of roof features in common areas of the surveys. Table 9 shows the different realignment times required for the adopted and compared point clouds.

**Table 9.** Approximated realignment time for the four different methodologies.

| Point Cloud | Realignment Time [min] |
|---|---|
| $\wp_{Markers}$ | 20 |
| $\wp_{ICP}$ | 120 |
| $\wp_{PARF}$ | 30 |

## 4. Conclusions

This paper presents a new and innovative methodology for the accurate realignment of large point clouds. The proposed technique is based on a planar approximation of geometric features associated with roofs (PARF); roofs can be recognized in most urban and archaeological environments. A new representation of the ideal flat feature associated with the roofs is proposed considering the roofs' size in the observation equation system. Specific experimentation proposed in the paper demonstrated their low sensitivity to noise compared to other analyzed methodologies. For these reasons, by the proposed methodology, roofs can represent a strong reference element for determining the relative realignment of point clouds. In order to evaluate the accuracy, the presented methodology has been compared with other very spread methods, which are the marker-based and ICP realignment. The results evidence that the PARF-based method is better than the marker-based one and comparable with ICP, although significantly less expansive in computational terms with respect to this one. The proposed representation of the ideal flat features associated with the roofs allows for an optimization that better considers the spatial distribution of roofs and not simply their spatial position and orientation. A significant advantage of the methodology presented here is that it does not require the use of any

introduced element in the surveyed area. Therefore, a PARF-based realignment can be performed even after years of respect to a marker-based one. Moreover, this methodology also has a low sensitivity with respect to environmental changes. For example, this is not verified for the ICP, where the season changes may lead to significant variations (leaves falling from the trees, weeds, etc.) in the area, preventing the algorithm from converging. At the same time, the ICP requires an excellent initial coarse alignment to converge, differently from the PARF. The proposed method requires less computational resources than the ICP, converges faster, and has not required the decimation of the point cloud, which also requires a lot of time to be performed. The presented methodology requires isolating a tiny part of the point cloud, resulting in more efficiency than other feature-based methodologies. Given the promising results obtained by the proposed association and registration methods, further efforts will be spent on automating the only manual processing: roof segmentation and recognition.

**Author Contributions:** Conceptualization, L.D.A., P.D.S. and E.G.; methodology, L.D.A., P.D.S. and E.G.; software, L.D.A. and E.G.; validation, M.A., E.G. and S.Z.; investigation, E.G.; resources, M.A., D.D. and S.Z.; data curation, E.G.; writing—original draft preparation, L.D.A. and E.G.; writing—review and editing, M.A., P.D.S., E.G. and S.Z.; visualization, M.A., E.G. and S.Z.; supervision, P.D.S. and D.D. All authors have read and agreed to the published version of the manuscript.

**Funding:** This research received no external funding.

**Data Availability Statement:** Not applicable.

**Conflicts of Interest:** The authors declare no conflict of interest.

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
