# Peer review of "Fast and Accurate Registration of Terrestrial Point Clouds Using a Planar Approximation of Roof Features"

_remotesensing, doi:10.3390/rs14132986_

Round 1

Reviewer 1 Report

This paper presents a methodology for the accurate realignment of large point clouds based on a planar approximation of geometric features associated with roofs (PARF); Specific experimentation proposed in the paper demonstrated their low sensitivity to noise compared to other analyzed methodologies. However, this manuscript is not suitable for acceptance due to follows:

1. The innovation of the method is not listed clearly in the paper

2. The proposed method is not fully automated in the key problem of point cloud alignment:

1)The overlapping part of two point clouds needs to be judged manually.

2)Alignment primitives - roofs need to be manually segmented.

3)Finding the corresponding Ideal features is very important in the registration task, and we did not find an clear introduction in the paper. If done manually, it will also reduce the automation of the method.

4)How to get the reference point is not clear, the results are likely to be manually selected according to Figure4 and Figure8.

3. The elements such as Roof and building facade in urban can be associated with plane primitives and have been widely used in the field of point cloud processing, which is a common method for researchers. An experimentation is carried out in Seciton2.1 which does not reflect innovation or discovery. Moreover, it is not clear that how to calculate planar parameters in this paper.

4. From the experimental results:

1) compared with the marker-based method, although the accuracy of the proposed is improved, the procedure requires more manual processing;

2) compared with classical ICP, although the proposed does not require down-sampling and excellent initial coarse alignment, the accuracy does not outperform it, and ICP is fully automatic.

3) In addition, “the proposed method doesn’t need a good initial alignment” is not exhibited and explained in the experiment.

5. Others:

1)Line248: ‘fixed point cloud’ and ‘moving one’ is denoted as the same?

2)Line374:”Section??”

Author Response

The answers to the reviewer #1 are reported in the attached file 

Reviewer 2 Report

Page1, Line 11:

»the use of GCPs, and other ones, aligned by ICP«

You need to provide the meaning of these abbreviations at the first mention for the reader that are not familiar with them.

Page 4, Line 173:

 “that h non-ideal features are identified”

What is “h”? The number of these non-ideal features?

Page 9, Line 272:

“a suitable solution has been found using an appropriate research space in all the considered scenarios”

What is “appropriate” research space? For the ICP could be a good initial (coarse) alignment.

Page 11, Line 296:

“The acquired images have been processed into Agisoft Metashape Pro, obtaining two different point clouds”

As I understand, the point clouds are artificially generated from the acquired images? No LiDAR is used? Is your approach applicable also for the clouds fetched by the LiDAR systems?

Page 14, Table 8:

It would be better if you present error as an absolute difference that in percentage (and also calculate and presents the average error in Table 8).

Page 15, Line 374:

“quantitatively experimented in Section ??.”

The section number is missing.

Author Response

The answers to reviewer #2 are reported in the attached file 
